

# Identifying meteorological drivers of extreme impacts: an application to simulated crop yields

Johannes Vogel[1,2,*], Pauline Rivoire[3,4,*], Cristina Deidda[5], Leila Rahimi[5,6], Christoph Alexander Sauter[7], Elisabeth Tschumi[3,8], Karin van der Wiel[9], Tianyi Zhang[10], and Jakob Zscheischler[3,8]

[1]Institute of Environmental Science and Geography, University of Potsdam, Potsdam, Germany
[2]Institute of Ecology, Technical University of Berlin, Berlin, Germany
[3]Oeschger Centre for Climate Change Research, University of Bern, Bern, Switzerland
[4]Institute of Geography, University of Bern, Bern, Switzerland
[5]Department of Civil and Environmental Engineering, Politecnico di Milano, Milano, Italy
[6]Department of Water Engineering, University of Tabriz, Iran
[7]Department of Civil and Environmental Engineering, University of Strathclyde, Glasgow, United Kingdom
[8]Climate and Environmental Physics, University of Bern, Bern, Switzerland
[9]Royal Netherlands Meteorological Institute (KNMI), De Bilt, the Netherlands
[10]Institute of Atmospheric Physics, Chinese Academy of Sciences, Beijing, China
[*]These authors contributed equally to this work.

**Correspondence:** Johannes Vogel (joschavogel@uni-potsdam.de), Pauline Rivoire (pauline.rivoire@giub.unibe.ch), Tianyi Zhang (zhangty@post.iap.ac.cn)

**Abstract.** Compound weather events may lead to extreme impacts that can affect many aspects of society including agriculture. Identifying the underlying mechanisms that cause extreme impacts, such as crop failure, is of crucial importance to improve understanding and forecasting. In this study we investigate whether key meteorological drivers of extreme impacts can be identified using Least Absolute Shrinkage and Selection Operator (Lasso) in a model environment, a method that allows for

automated variable selection and is able to handle collinearity between variables. As an example of an extreme impact, we investigate crop failure using annual wheat yield as simulated by the APSIM crop model driven by 1600 years of daily weather data from a global climate model (EC-Earth) under present-day conditions for the Northern Hemisphere. We then apply the logistic Lasso regression to predict which weather conditions during the growing season lead to crop failure. We obtain good model performance in Central Europe and the eastern half of the United States, while crop failure years in regions in Asia

and the western half of the United States are less accurately predicted. Model performance correlates strongly with annual mean and variability of crop yields, that is, model performance is highest in regions with relatively large annual crop yield mean and variability. Overall, for nearly all grid points the inclusion of temperature, precipitation and vapour pressure deficit is key to predict crop failure. In addition, meteorological predictors during all seasons are required for a good prediction. These results illustrate the omnipresence of compounding effects both between meteorological drivers and different periods

of the growing season for creating crop failure events. Especially vapour pressure deficit and climate extreme indicators such as diurnal temperature range and the number of frost days are selected by the statistical model as relevant predictors for crop failure at most grid points, underlining their overarching relevance. We conclude that the Lasso regression model is a useful tool to automatically detect compound drivers of extreme impacts, and could be applied to other weather impacts such as wildfires





or floods. As the detected relationships are of purely correlative nature, more detailed analyses are required to establish the

causal structure between drivers and impacts.

## 1  Introduction

Climate extremes such as droughts, heatwaves, floods and frost events can have substantial impacts on crop health (Shah and Paulsen, 2003; Singh et al., 2011; Lesk et al., 2016; Ben-Ari et al., 2018). However, not all climate extremes lead to an extreme impact, and large impacts can be related to moderate drivers (Zscheischler et al., 2016; Van der Wiel et al., 2019a, 2020; Pan

et al., 2020). Whether a large impact occurs does not only depend on a climate hazard but also on the vulnerability of the underlying system (Oppenheimer et al., 2015), which varies strongly for crops during the course of the growing season (Iizumi and Ramankutty, 2015; Ben-Ari et al., 2018). The mechanisms that translate a climate hazard into crop failure are often very complex and associated with lagged effects that are difficult to disentangle (Frank et al., 2015).

While climate extremes may lead to large impacts, extreme climate-related impacts are often the result of multiple contribut-

ing factors. The concept of compound events has recently been promoted to address climate impacts from an impact-centred perspective. For instance, compound events have been defined as extreme impacts that depends on multiple statistically dependent drivers (Leonard et al., 2014) or, more recently, simply as the combination of multiple drivers that contributes to environmental or societal risk (Zscheischler et al., 2018). Drivers in this context refer to climate and weather processes and phenomena. With respect to yields at the local scale, multiple drivers can compound an impact through a sequence of weather

events (temporally compounding); one weather event may also change the vulnerability of the crop to a subsequent weather event (preconditioning); or multiple drivers may interact and impact crops at the same time (multivariate events) (Zscheischler et al., 2020).

Understanding the drivers that lead to extreme impacts helps to better predict and mitigate the potential impacts of such events. One way of identifying the relevant drivers of an impact is to perform a bottom-up analysis, that is, start from an

impact and identify key drivers through statistical analysis (Zscheischler et al., 2013; Ben-Ari et al., 2018). In this context, linear regression analysis can identify the most relevant drivers of an impact variable and reveal potential interactions between drivers (Forkel et al., 2012; Ben-Ari et al., 2018). More sophisticated approaches such as random forest might yield higher predictive power at the cost of losing explainability (Vogel et al., 2019). When the set of possible predictors is very large, suitable variable selection approaches need to be applied to reduce the number of predictors. In order to be applicable to

a large number of locations and a variety of impacts, an automatic approach is desired that only requires a limited amount of expert knowledge and parameter tuning. An example of such an approach is the Least Absolute Shrinkage and Selection Operator (Tibshirani, 1996), or short Lasso regression, which obtains a reduced number of predictors by penalizing the number of variables in the loss function.

The aim of this study is to present a method that can identify drivers of extreme impacts in an automatic manner and

that is suitable for many applications. We use crop failure as an example of an extreme impact in a model environment, that is, we use simulated data from a climate and a crop model. End-of-season crop yield is related to climate drivers via



highly complex interactions at different temporal scales. Temperature and precipitation are the two basic climate variables that regulate crop health (Lobell and Asner, 2003; Lobell et al., 2011; Leng et al., 2016). Furthermore, vapour pressure deficit (VPD), the difference of water vapour pressure at saturated condition and its actual value at a given temperature, determines

crop photosynthesis and water demand (Rawson et al., 1977; Zhang et al., 2017; Yuan et al., 2019).

Here we use 1600 years of wheat yield data from a global gridded crop model driven by simulated meteorological data under present-day conditions. Based on this large database of yield data we showcase approaches to identify multiple drivers of crop failure in different regions of the world and highlight results for the Lasso regression. Using a model environment to explore new analytical approaches to identify drivers of extreme impacts, we circumvent common limitations associated

with observational data, such as a small sample size, measurement uncertainties and data coverage. Among the large amount of information provided by the crop model simulations, the statistical model summarizes the link between crop failure and climate conditions.

This paper is structured as follows. The data and methods used in this study are introduced in section 2. In this section, the reader can first find a description of the data, including an introduction to the global climate model and the crop model

used in this study. We further describe which meteorological variables are considered in the statistical analysis; section 2 also introduces the Lasso logistic regression to predict years of low yield based on meteorological drivers and the metrics employed to assess the performance of the statistical model. The results of the Lasso regression are shown in section 3, where the performance and the summary statistics for the variables that have been selected as being critical to predict crop failure events are presented. Finally, we summarize and discuss the Lasso regression's results in section 4, and give some perspective

to this study.

## 2  Data and Methods

### 2.1  Climate and crop model simulations

The EC-Earth global climate model (v2.3, Hazeleger et al., 2012) was used to create 2000 years of present-day weather simulations. Present-day was defined as the five year model period in which the global mean surface temperature matched that

observed in 2011-2015 (HadCRUT4 data, Morice et al., 2012). Because of a cold bias in EC-Earth, in the model this period is 2035-2039. To create the large ensemble, twenty five ensemble members were branched off from sixteen long transient climate runs (forced by Representative Concentration Pathway (RCP)8.5). Each ensemble member was integrated for five years. Differences between ensemble members were forced by choosing different seeds in the atmospheric stochastics perturbations (Buizza et al., 1999). In total there are $16 \times 25 \times 5 = 2000$ years of meteorological data, at T159 horizontal resolution (approximately

$1°$). More details on these climate simulations are provided in Van der Wiel et al. (2019b).

Biases in the EC-Earth simulations result in unrealistic growing conditions for crops. Therefore, minimum and maximum temperatures and precipitation fields were bias corrected. The AgMERRA reanalysis (Ruane et al., 2015) was used as 'truth'. From AgMERRA the years 1981-2010 were used as a training set, while EC-Earth uses the long transient runs (sixteen × 2005-2034). Daily minimum and maximum temperatures were corrected on a grid point basis, a model bias field was defined





as the difference between the model climatology and the AgMERRA climatology. The climatology was defined to be the mean plus the first three annual harmonics. Daily precipitation was corrected towards having the correct number of rainy days and total amount of precipitation. Firstly, for each month the number of rainy days in AgMERRA was computed (threshold 0.1 mm/day), then the same threshold was determined for EC-Earth data, which resulted in the same number of rainy days. All days with simulated precipitation smaller than this threshold were set to 0 mm/day. Lastly, the total amount of precipitation

was corrected by means of a multiplicative factor, also on a month-by-month basis. Other meteorological variables were not bias corrected.

Northern Hemisphere winter wheat yields were simulated using the APSIM-Wheat model (Zheng et al., 2014), which is a process-based model incorporating wheat physiology, water and nitrogen processes under a wide range of growing conditions. It was previously used for field (Li et al., 2014), regional (Asseng et al., 2013) and global scale (Rosenzweig et al., 2014) wheat

studies. A grid point-specific sowing date was used based on Sacks et al. (2010). The application of nitrogen was exacted from Mueller et al. (2012). Soil parameters (including pH, soil total nitrogen, organic carbon content, bulk density and soil moisture characteristics curves for each of five 20 cm deep soil layers) were derived from the International Soil Profile Dataset (Batjes, 2012). In addition, we also input the grid-specific thermal time accumulation parameters, which were derived from phenology (Sacks et al., 2010) and AgMERRA data. The atmospheric $CO_2$ concentration was set to 394 ppm. The growing season of

winter wheat spans two calendar years (e.g. sowing in November and harvest in June). As such each climate model integration of five years covers four winter wheat growing seasons, the 2000 years of EC-Earth climate data thus result in 1600 simulated wheat growing seasons.

## 2.2  Data processing

The APSIM model provided crop data for 995 grid points in the Northern Hemisphere. For our analysis, we chose to discard

all grid points below the 10[th] percentile of annual mean crop yield because many of these grid points were also associated with unrealistically long (>365 days) or short (<90 days) growing seasons or had an overall average crop yield of 0 kg/ha. 895 grid points remained for the analysis.

At each grid point, a year with yield lower than the 5[th] percentile for this grid point is considered as a year with crop failure, and called "bad year" in the remainder, whereas all other years are referred to as "normal years". Grid points for which the 5[th]

percentile yield was equal to 0 were excluded to avoid the co-occurrence of years without yield in the bad and normal years. This excluded 6 more grid points so that 889 remained for further analysis. Figure 1 shows the simulated mean annual yield and indicates grid points that were discarded for further analysis. Finally, we discarded individual years with a growing season longer than 365 days, leading to a slightly smaller number of years than 1600 for 82 pixels, i.e. for about 5 % of the grid points.

The data was split into a training and testing dataset by randomly assigning 70 % of the data to the former and 30 % to the

latter. For the logistic regression (Section 2.4) explanatory variables and yield were normalised by rescaling them to a range of [-1, 1] for each grid point individually.





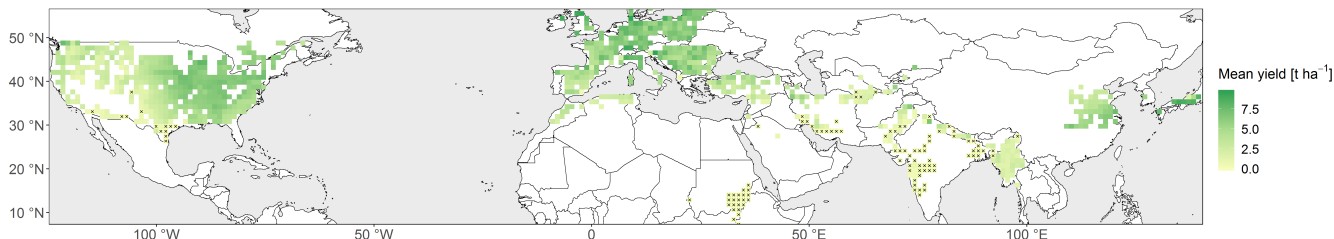

**Figure 1.** Mean annual yield over the 1600 years (tonne/hectare). Grid points discarded for our study are crossed out (specified in the Section 2.2).

## 2.3   Explanatory data analysis

The APSIM model uses six meteorological variables on a daily basis as input (dew point temperature ($T_d$), precipitation (Pr), 10 m wind speed (Wind), incoming shortwave radiation (Rad), maximum temperature ($T_{max}$), and minimum temperature ($T_{min}$)). From these variables, we additionally calculated vapour pressure deficit (VPD) as an important variable for plant growth (Rawson et al., 1977; Zhang et al., 2017; Yuan et al., 2019). Figure 2 illustrates the temporal evolution of composites of these seven variables over the course of a growing season for normal (blue) and bad years (red) for one grid point in France (47.7° N, 1.1° E, Fig. 2a). The composites provide some indication about which of the meteorological variables may contribute to crop failure. In addition, the temporal evolution of the two composites reveals during which part of the growing season the different variables are relevant. The various composites suggests that, for this grid point, 30-day Pr, VPD and $T_{max}$ during the summer (June-August) have a high impact on crop yield (Figs. 2c, f and h). The other variables appear to be less relevant (Figs. 2b, d, e and g). Similar composites for grid points in the US (90.0° W, 44.3° N) and in China (118.1° E, 30.8° N) are shown in Figs. A1 and A2, respectively.

In addition to the seven meteorological variables, we considered seven climate extreme indicators as potential predictors of crop failure (mean diurnal temperature range, dtr; number of frost days, frs; maximum temperature, TXx; minimum temperature, TNn; maximum five day precipitation sum, Rx5day; number of warm days, TX90p; number of cold days, TN10p; following Vogel et al., 2019) (Table 1). For both the monthly means of the meteorological variables, as well as for the growing season means/totals of the indicators of climate extremes we calculated the Pearson correlation coefficient between the variables and annual yield (Figs. 3a and b for the same grid point as in Fig. 2 and Figs. 3c and d as average correlation over all grid points). These correlations are computationally and conceptually very simple and together with Fig. 2 serve as a first estimation of the importance of the available variables. Some variables, such as wind speed, do not have a discernible influence on yield and thus can be neglected for this study. We use monthly means of $T_{max}$, Pr and VPD, as well as the seven extreme indicators for further analysis.

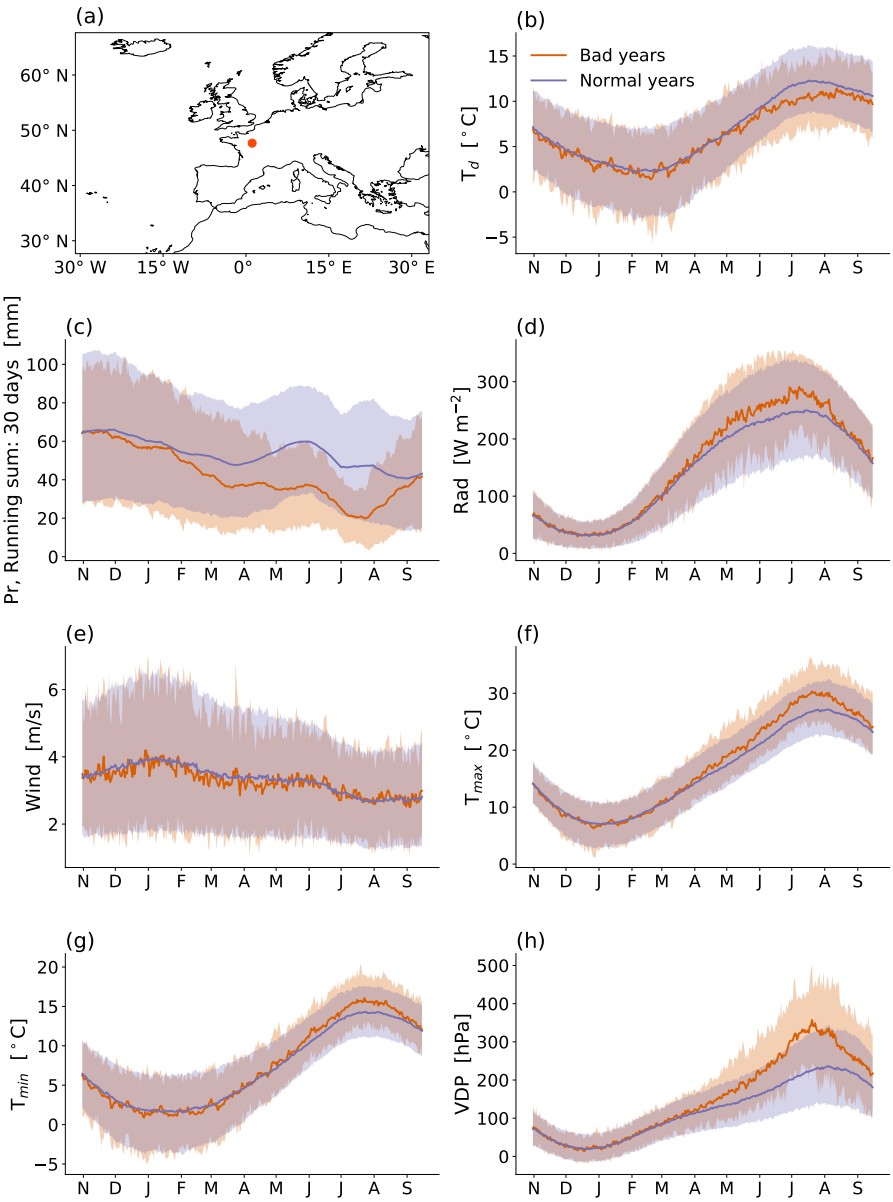

**Figure 2.** Daily evolution of meteorological variables used as input for the APSIM model over the course of the year for an exemplary grid point in France (47.7° N, 1.1° E, location indicated in (a)). Red lines indicate the composite mean of the bad years (80 seasons), blue lines the composite mean of the normal years (1520 seasons). Shading shows the range between the $10^{th}$ and $90^{th}$ percentile of the respective years. Variables shown are (b) dewpoint temperature, (c) 30-day running sum of precipitation, (d) incoming shortwave radiation, (e) wind speed, (f) maximum temperature, (g) minimum temperature, and (h) vapour pressure deficit (VPD).





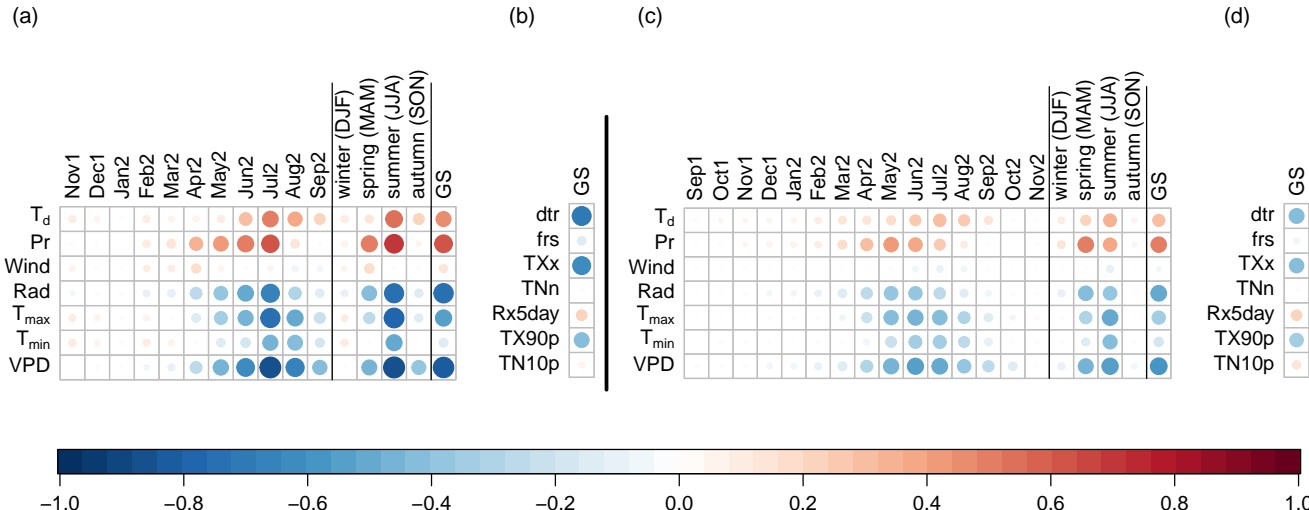

**Figure 3.** Linear correlations between potential meteorological predictors and annual yield. (a) Correlation between the monthly, seasonal and growing season (GS) averages of the meteorological variables and annual yield for a grid point in France (47.7° N, 1.1° E). (b) Correlation of the climate extreme indicators (Table 1) and annual yield for the same grid point. (c, d) Average of the same correlations across all Northern Hemisphere grid points.

**Table 1.** Meteorological drivers used in the analysis

| Variable name | Description | Type |
|---|---|---|
| $T_{max}$ | Maximum temperature | Monthly mean |
| VPD | Vapour-pressure deficit | Monthly mean |
| Pr | Precipitation | Monthly mean |
| dtr | Mean diurnal temperature range in the growing season | Climate extreme indicator |
| frs | Number of frost days in the growing season | Climate extreme indicator |
| TXx | Maximum temperature in the growing season | Climate extreme indicator |
| TNn | Minimum temperature in the growing season | Climate extreme indicator |
| Rx5day | Maximum five day precipitation sum in the growing season | Climate extreme indicator |
| TX90p | Number of warm days in the growing season with daily maximum temperature above the 90[th] percentile [a] | Climate extreme indicator |
| TN10p | Number of cold days in the growing season with daily minimum temperature below the 10[th] percentile [a] | Climate extreme indicator |

[a] Note: Percentiles are grid point based, i.e. they are representative for the local climate





### 2.4 Lasso regression

The aim of this study is to provide an interpretable statistical model able to predict years with extremely low yields (bad years) with meteorological variables. We use the Least Absolute Shrinkage and Selection Operator (Lasso, Tibshirani, 1996) logistic regression for an automatic selection of meteorological variables that are statistically linked to low yields. The approach is the following.

For a given grid point, let $Y \in \{0,1\}^n$ be the binary yield vector, with $n$ the number of years. If the year $i \in \{1, \ldots, n\}$ is a
bad year, then $Y_i = 0$, otherwise $Y_i = 1$. Let $X_1, \ldots, X_p \in \mathbb{R}^n$ be the explanatory variables vectors (monthly meteorological variables and climate extreme indicators, rescaled as explained in Section 2.2). Using a generalized linear model and, more specifically, a logistic regression, we can identify how much of the occurrence of bad yields is explained by which explanatory variable:

$$\mathbb{P}[Y = 0] = \frac{1}{1 + \exp(\beta_0 + \beta_1 \times X_1 + \cdots + \beta_p \times X_p)} \qquad (1)$$

where $\beta_0, \beta_1, \ldots, \beta_p$ are the regression coefficients.

However, a simple logistic regression presents two challenges here. Firstly, some variables might be highly correlated (auto-correlation of meteorological variables, e.g. correlation between temperature in May and temperature in June, or the correlation of extreme indices with meteorological variables). This correlation implies a high variability of the coefficients. For instance, if the variables $X_j$ and $X_k$ are highly correlated, the information brought by a high absolute value of $\beta_j$ and a low absolute value
of $\beta_k$ might be the same as the information brought by a low absolute value of $\beta_j$ and a high absolute value of $\beta_k$. Another issue is the large number of potential explanatory variables (up to 43 for some grid points). The relatively straightforward relationship of a generalized linear model (simpler than the crop model equations themselves) allows to reveal which meteorological variables explain bad yields best. However, if the number of *a priori* explanatory variables is very large, the regression becomes rather complex and many coefficients will be close to zero, rendering an interpretation difficult.

Lasso regression tackles both challenges with an automatic variable selection using a regularization by penalizing the number of coefficients different from 0 using the $\ell_1$ norm on the vector of coefficients (Tibshirani, 1996). Thus, the regression coefficients are obtained by minimizing an objective function consisting of the sum of the usual loss function for logistic regression and a penalty term on the coefficient norm:

$$\min_{(\beta_0, \beta) \in \mathbb{R}^{p+1}} -\left[ \frac{1}{n} \sum_{i=1}^{n} y_i \cdot (\beta_0 + x_i^T \beta) - \log(1 + e^{\beta_0 + x_i^T \beta}) \right] + \lambda ||\beta||_1, \qquad (2)$$

for a fixed $\lambda > 0$. The penalty term on the coefficient norms prevents a high variability of these coefficients. Furthermore, the $\ell_1$ norm implies a variable selection. Coefficients associated to non-relevant explanatory variables are set to 0.

We use the R package `glmnet` (Friedman et al., 2010) to perform the Lasso regression with R version 3.6 (R Core Team, 2019). Through 10-fold cross-validation in the training dataset, we obtain the optimal $\lambda_{min}$ and $\lambda_{1se} = \lambda_{min} + se$ with $se$ the standard error of the lambda that achieves the minimum loss, and the coefficients $\beta$, which are the solution to the optimization
in equation (2) for $\lambda = \lambda_{1se}$. Our preference for $\lambda = \lambda_{1se}$ is motivated by the balance between number of selected variables and





accuracy of the loss function minimization. Indeed, less variables are selected with $\lambda_{1se}$ than with $\lambda_{min}$, because $\lambda_{1se} > \lambda_{min}$ and thus the penalty term on the norm of coefficient is stronger, but the minimization of the equation (2) is still sensible, because $\lambda_{1se}$ lies within the uncertainty range of the optimal $\lambda$.

## 2.5 Other models

To compare the performance of the Lasso regression with other regression methods we also perform the analysis with a Generalized linear model (GLM) and a random forest binary classification.

For the application of the GLM, a pre-selection of the initial variables is required, since the number of predictors is limited. Only the variables with the highest Pearson correlation coefficient ($\rho > 0.30$) were selected as initial predictors from an initial dataset composed by all months of the growing season for each of the three variables ($T_{max}$, Pr and VPD) and the seven extreme

indicators. Next, the subset of best predictor variables is identified with the leaps algorithm (Furnival and Wilson, 1974). We use the implementation of the R package `bestGLM` (McLeod et al., 2020). Overall, GLM achieves lower performance (Section 2.7) compared to the Lasso logistic regression (not shown). The weaknesses of this approach is its sensitivity to outliers and multicollinearity, and overfitting.

Finally, a random forest approach – a common machine learning technique – was also performed using the R package

`randomForest` (Breiman, 2001; Liaw and Wiener, 2002) serving as a benchmark for the model performance of the Lasso logistic regression. The random forest binary classification achieves comparable performance (Section 2.7) but is not superior to the Lasso approach.

## 2.6 Cut-off value adjustment

The cut-off value for assigning a continuous prediction to either a bad or normal year was adjusted grid point-wise to account

for the unbalanced data set with 19-fold higher occurrences of normal years than bad years. Let $s$ be the segregation threshold between bad year predicted and good year predicted ($s$ is the local cut-off value for each grid point). In other words, if the probability $p = \mathbb{P}[Y = 1]$ predicted for a given grid point by the Lasso logistic regression model is greater or equal to $s$ (resp. lower than $s$), then the year is predicated as a normal year (resp. bad year). We want to choose $s$ as a good compromise in prediction of normal years and bad years, given that bad years are rare. In other words, we want to find an optimal trade-off

between specificity and sensitivity. To this purpose, a cost function $\mathcal{C} = \mathcal{C}(s)$ is calculated based on the false positive rate $R_{FP} = R_{FP}(s)$, the associated cost for a false positive instance $\mathcal{C}_{FP}$, the sum of observed bad years $O_{BY}$, the false negative rate $R_{FN} = R_{FN}(s)$, the associated cost for a false negative instance $\mathcal{C}_{FN}$ and the sum of observed normal years $O_{NY}$ of the training data set (Hand, 2009). A false positive means that a bad year was observed while a normal year was predicted, and a false negative refers to the observation of a normal year, whereas a bad year was predicted. For a given grid point, FP, FN, TP and


TN denote the total number of false positives, false negatives, true positives and true negatives, respectively (Fig. 4). The value of $\mathcal{C}(s)$ is given by:

$$\mathcal{C}(s) = R_{\mathrm{FP}}(s)\mathcal{C}_{\mathrm{FP}}O_{\mathrm{BY}} + R_{\mathrm{FN}}(s)\mathcal{C}_{\mathrm{FN}}O_{\mathrm{NY}}, \tag{3}$$

where $R_{\mathrm{FP}} = \dfrac{\mathrm{FP}}{\mathrm{FP}+\mathrm{TN}}$, $R_{\mathrm{FN}} = \dfrac{\mathrm{FN}}{\mathrm{FN}+\mathrm{TP}}$ and $\mathcal{C}_{\mathrm{FP}} = \mathcal{C}_{\mathrm{FN}} = 100$. In this study, the cost associated with false positive $\mathcal{C}_{\mathrm{FP}}$ and false negatives $\mathcal{C}_{\mathrm{FN}}$ are given equal weight.

The optimal segregation threshold $s^*$ for a given grid point is $s^* = \mathrm{argmin}_{s\in(0,1)}\mathcal{C}(s)$. The cut-off level selected in this study is the mean value of $s^*$ over all grid points.

Observed

| | Normal year $(Y = 1)$ | Bad year $(Y = 0)$ |
|---|---|---|
| Predicted — Normal year $(Y = 1)$ | TP | FP |
| Predicted — Bad year $(Y = 0)$ | FN | TN |

**Figure 4.** Confusion matrix for classification of observed and predicted normal and bad years.

## 2.7  Model performance assessment and sensitivity analysis

Model performance is assessed using the critical success index (CSI). The CSI is frequently used for evaluating the prediction of rare events, as it neglects the number of correct predictions of non-extremes, which dominate the confusion matrix (Mason,
1989). General performance measures such as the misclassification error are biased by the high number of normal years and are therefore not meaningful for the assessment of model performance in unbalanced datasets with underrepresented extreme events. The CSI is defined as

$$\mathrm{CSI} = \frac{\mathrm{TN}}{\mathrm{TN}+\mathrm{FP}+\mathrm{FN}}. \tag{4}$$

To evaluate the robustness of our model, in addition to the 5th percentile threshold we repeated the analysis with segregation
thresholds of 2.5 % and 10 %, reaching qualitatively similar performance. Additionally, we applied two more combinations of splitting training and testing data set, a 60/40 and 80/20 split. With increasing size of the training data set, the CSI increased slightly, however at the expense of stochastic under-representation of bad yield years in the smaller testing data sets. As a trade-off, we decided for the 70/30 split.



We further explored whether accounting for first order interaction terms between variables would increase performance.
This was done using the R package `glinternet` (Lim and Hastie, 2015, 2019). Overall, inclusion of interaction terms did
not improve model performance. Because incorporating interaction terms increases model complexity considerably while only
contributing little to model performance, they were not included in our final model.

The adjustment of the cutoff value was carried out with equal weight to false positive and false negative predictions. It can
be argued that the latter case – where a normal year is predicted, but crop failure is observed – is more detrimental and should
therefore be given a higher weight. Due to the subjectivity in the determination of this weight, an adjustment of the weight
term was not applied. However, it should be noted that the attribution of a higher weight of false negative predictions would
yield a higher cutoff value and hence improve the overall CSI.

## 3   Results

### 3.1   Overall performance

The Lasso logistic regression model can predict bad years with an average $CSI = 0.43$ across all grid points. Best performance
is obtained in the eastern half of the United States with a maximum of $CSI = 0.82$ (Fig. 5), which decreases eastward in the
Great Plains and are lowest in the wheat growing regions located close to the Rocky Mountains. Furthermore, especially the
most northern and southwestern grid points in North America show generally lower performance. Also central Europe shows
high performances up to $CSI = 0.80$. A notable regional exception with low performances can be found in the Alps. Many Asian
and African growing regions show medium prediction accuracy such as northern China, Myanmar, Turkey and the Maghreb,
with exceptions of some regions including Pakistan, southern China and Japan, which show generally low performance. For
30 grid points, it is not possible to obtain reasonable predictions of bad years with our approach, indicated by a CSI equal to
0. Overall, regions with high prediction accuracy of bad years are often those that also have high mean yields (Fig. 1). CSI is
positively correlated with mean yield with a Pearson's correlation coefficient of $\rho = 0.46$ (Fig. 6a), an even stronger correlation
is found with yield variability ($\rho = 0.57$) (Fig. 6b).

### 3.2   Explanatory variables

Here we summarize properties of the variables selected by Lasso logistic regression as relevant explanatory variables, i.e.,
which are statistically linked to bad years. A median of 11 variables per grid point has been selected as explanatory variables,
and for 50 % of grid points the number of selected variables lies between 7 and 14 (Fig. 7a). The inclusion of extreme indicators
provides a useful addition to the monthly predictors, shown by a median number of two selected extreme indicators per grid
point (Fig. 7b). Grid points without extreme indicators are found only in few areas such as eastern Europe, the Alps and
southern China. 72 % of all grid points include monthly predictors of VPD, Pr and $T_{max}$ and almost all grid points (97 %)
incorporate VPD (Fig. 7c). Interestingly, in the Great Plains (USA) in many cases temperature is not included, whereas in most
other regions of the USA all meteorological variables are selected to achieve a good prediction. In southern China, temperature

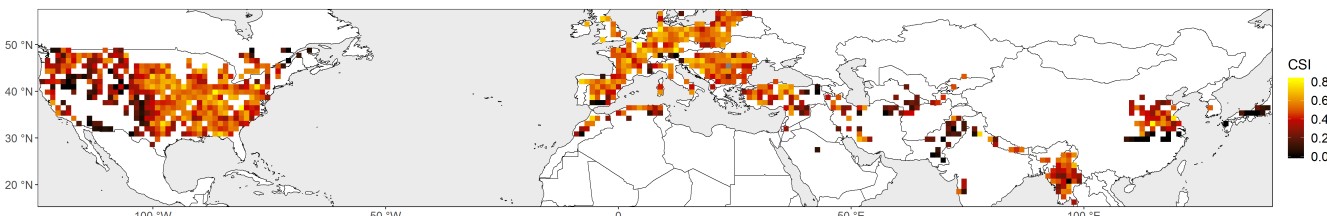

**Figure 5.** Critical success index (CSI, equation (4)) of the Lasso logistic regression model. (See Section 2.7 for definition).

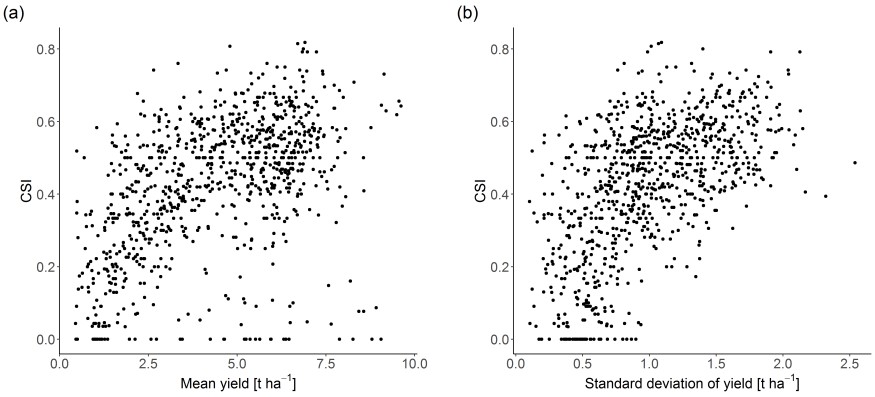

**Figure 6.** Correlation between Critical Success Index (CSI) and annual crop yield mean and variability for the 889 pixels included in the Lasso logistic regression model. (a) Scatterplot between CSI and mean annual yield. (b) Scatterplot between CSI and annual yield standard deviation.

is not needed by the models, whereas in the northern areas, usually all meteorological variables are part of the model. In most wheat growing regions, particularly in northeastern USA, southeastern Europe and Turkey, all four seasons contain relevant predictors for predicting bad years (Fig 7d). Generally, the number of seasons included decreases towards the southeastern regions in the USA, whereas in western Europe no clear pattern emerges. In lower latitudes such as southern Asia, growing seasons are generally shorter (Fig. A3) and consequently often only predictors from one or two seasons are included in the
respective models.

At the global scale, VPD in May and June, as well as Pr in April are the predictors which are most often included in the Lasso regression, followed by the climate extreme indicators diurnal temperature range (dtr) and number of frost days (frs) (Fig. 8a). In nearly all cases the sign of the coefficient is negative for VPD in May and June, and positive for Pr in April. This implicates that higher VPD increases the risk of crop failure, and similar for the other variables. In North America and Europe,
in addition to dtr and frs, VPD and Pr in spring to early summer are the most frequent monthly predictors (Fig. 8b, c). The growing season for wheat varies with latitude. Consequently, in more northern regions, mostly in Europe and North America,

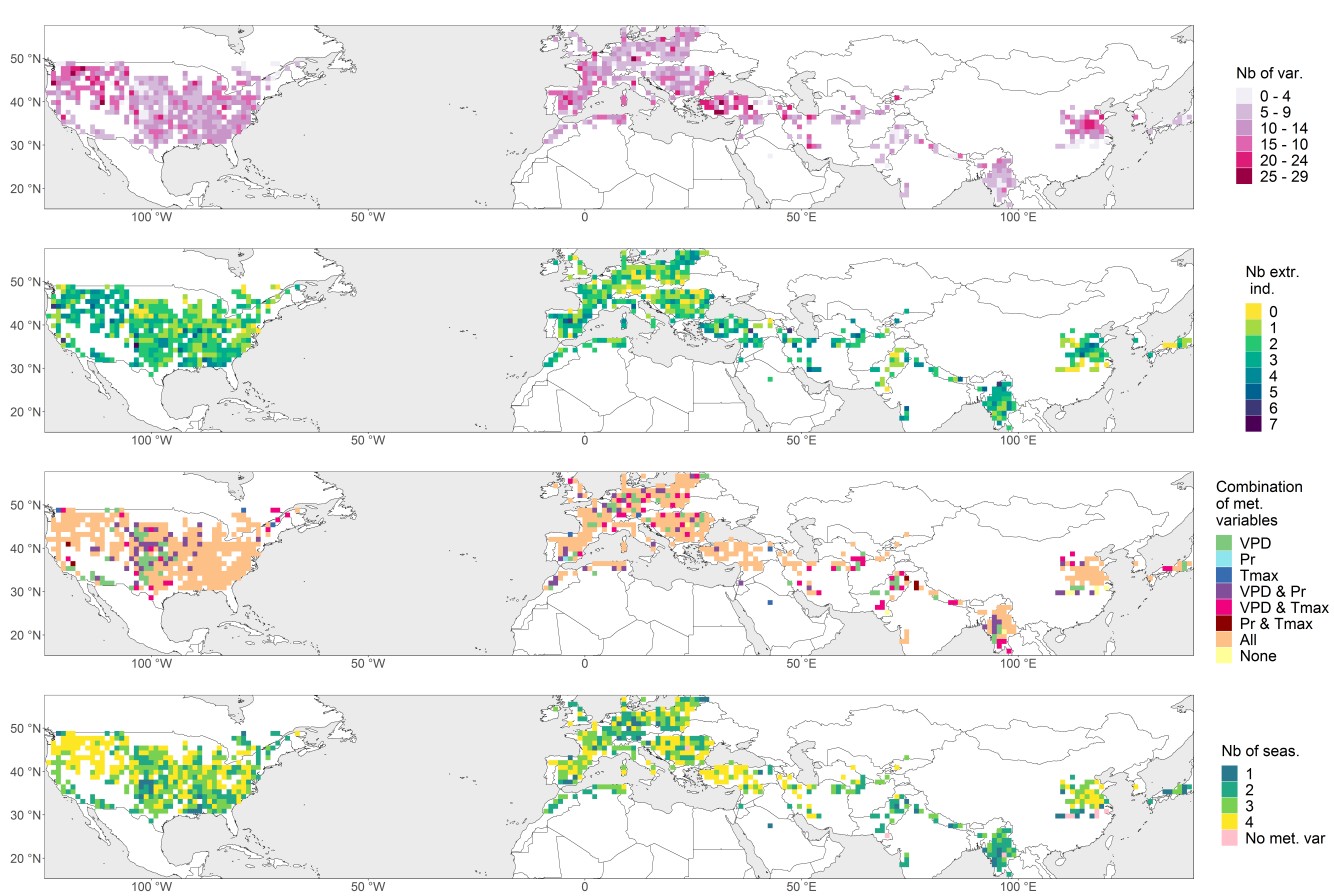

**Figure 7.** Maps illustrating the selected predictors by the Lasso logistical regression. (a) Total number of selected variables. (b) Number of selected climate extreme indicators. (c) Combination of selected meteorological variables. "None" means that only climate extreme indicators were selected, "All" means that at least one month from each of the three meteorological variables (VPD, Pr, $T_{max}$) is selected. (d) Number of selected seasons (out of the four seasons DJF, MAM, JJA, SON).





monthly predictors from the months between March and July are included in the Lasso regression, whereas in southern regions such as in Asia and Africa, November to May are usually the most frequent months (Fig. 8d).

This clear latitudinal shift can be visually identified in North America from February to July, especially for VPD (insert
link to supplementary GIFs here). In central Europe the growing season ends latest, thus VPD in August is usually included in the model. In addition to the most common climate extreme indicators dtr and frs, Rx5day and TXx are among the most frequent predictors in Asia and North America, respectively. Overall, frs is mostly included in northern grid points, with notable exceptions in western Europe (Fig. A4a) and mainly with a negative coefficient (higher frs leads to more crop failure events), which can likely be attributed to the influence of mild maritime climate in those regions. In contrast, dtr is important in most
Asian grid points and especially in western Europe and the Maghreb, whereas in the Pannonian Basin and Turkey it is a less common predictor (Fig. A4b). The coefficient associated with dtr in the Lasso regression is mainly negative, except in parts of India and Myanmar. Some variability in mean diurnal temperature range might be beneficial for regions close to the equator where the variability in diurnal temperature is usually low. Generally, dtr can influence wheat yield both ways depending on the growing region, e.g. a low dtr can be beneficial because of a reduced occurrence of frost days, whereas a higher dtr might
indicate a positive effect because of increased solar radiation (Lobell, 2007). TX90p is a common variable in low latitudes with a negative coefficient, especially in the southern USA and Myanmar (Fig. A4c). This indicates that in these regions physiological temperature thresholds are occasionally exceeded, making TX90p a crucial variable in these areas. Rx5day is predominant in the western USA, the western Mediterranean and central Asia (Fig. A4d), which are all growing regions with comparably low average annual precipitation.





**Figure 8.** For each possible predictor we show the percentage of grid points for which this predictors has a non-zero coefficient in the Lasso logistic regression. (a) all continents (889 grid points in total), (b) North America (419 grid points), (c) Europe (233 grid points) and (d) Asia (210 grid points). The extension "Y1" means that the respective month belongs to the first calendar year of the growing season, while "Y2" means it belongs to the second calendar year of the growing season.

## 4    Discussion

### 4.1    Predicting bad yield years

In this study, we presented a method for identifying drivers of extreme impacts using crop failure as an example. Such approaches are highly sought after to identify compound drivers of large impacts (Zscheischler et al., 2020). Our method allows to investigate potential drivers at a global scale using a highly automated scheme based on Lasso regression. The benefits of





Lasso regression include its usage for automatic variable selection, its consideration of correlation between explanatory vari-
ables and its performance. Moreover, the statistical model obtained provides a logistic linear relationship between crop failure
and selected variables, which is much simpler to interpret than the crop model equations themselves or results obtained with
more complex machine learning approaches.

We defined bad years as years where the annual crop yield is below the 5$^{\text{th}}$ percentile and were able to predict those years
by using the Lasso regression with an average CSI of 0.43. This means that on average, the sum of the numbers of false
positives and false negatives is slightly higher than the number of true negatives (or accurate predictions of bad years). Our
model performance is somewhat comparable to results from (Vogel et al., 2019), who were able to explain 46 % of variation
in spring wheat anomalies using a similar set of predictors based on a random forest algorithm. In our case, more sophisticated
machine learning regression models such as random forest did not yield better prediction skill, indicating that performance in
the current set-up using monthly predictors for a binary classification of bad years likely cannot be much improved. This is
probably also related to the fact that predicting extremes of a continuous variable is challenging because no natural separation
between extremes and non-extremes exists. Another challenge arises from the highly asymmetric distribution of observed bad
and normal years. Even though in our case the total amount of samples per grid point is relatively large (1600, because we
used simulated crop yield data) the number of observed bad years is only 80 and thus can still be considered fairly small. This
suggests that an application of the approach on real data might be even more challenging, as observational sample sizes are
much smaller. Furthermore, modelling winter wheat yield is particularly challenging due to its long growing season (Vogel
et al., 2019).

A limitation to our study design is the pre-selection of potential predictor variables. Here we used monthly mean values
and a number of climate extreme indicators. More flexible averaging time periods for the predictors might result in higher
prediction accuracy due to better overlap with sensitive periods of the impact variable. For instance, in our crop yield example
meteorological predictors need to coincide with the respective phenological development stage because their impact can vary
depending on the phenophase. Wheat for example is known to require wet conditions in the vegetative phase, however prefers
dry conditions during ripening (Seyfert, 1960). Therefore, the application of monthly meteorological predictors might be
insufficient for accurate matching of meteorological drivers to the respective phenological phases. We explored the option
of automatically generating optimal time periods for the meteorological predictors by maximizing the difference between
the composites between normal and bad years. For instance, 30-days cumulative precipitation differs between normal and bad
years starting in February until end of August for a grid point in France (Figure 2c), wheres VPD only differs over May through
September (Figure 2h). Composite plots for a grid point in the US and in China are shown in Figures A1 and A2, respectively.
However, deciding when the separation between normal and bad years is large enough to start and end the optimal time periods
is challenging and difficult to generalize and thus automate, which was the aim of our method design. Nevertheless, such a
well-tuned selection of predictors has the potential to improve the prediction of bad years significantly and should thus be
explored in future research.

We find a strong correlation of the yearly mean and standard deviation of annual yield with the Lasso regression performance
indicator CSI (Figure 6). Low model performance at grid points with low yield variability suggests that the distinction between





normal and bad years is challenging at these locations. Regions with high annual yield are found primarily in central Europe and the eastern half of the United States, which also represent the regions with highest model performances. In contrast, many regions in Asia generally have lower average yields and lower prediction skill of bad years. This could be related to a calibration bias in the crop model, leading to a better representation of wheat growing processes in regions where wheat reaches higher yields in the real world. A further explanation for this phenomenon could be that the crop model is primarily designed for crop

growth at typical environmental conditions, whereas growing regions with conditions at the edge of the ecological niche of wheat might be less well represented.

Our analysis was based on a time-series model, which is one of the three principal statistical methods used to link crop yield with weather conditions, along with cross section model and panel model, as explained in Shi et al. (2013). Collinearity between explanatory variables is a recurrent issue when using these methods (Shi et al., 2013), that we addressed with the Lasso

regression. One example is VPD and $T_{max}$, that might be highly correlated, but still might individually contribute relevant information because they have a different impact on the plant process, as explained in Kern et al. (2018). Lasso regression did not completely discard one of these two variables, despite their high correlation.

### 4.2 Important predictors

For most grid points, VPD is the most important monthly predictor of bad years, followed by Pr and $T_{max}$ in that order. While

their importance in time differs between grid points, depending on the timing of the respective growing season (Sippel et al., 2016), their order changes little across space. In addition, the order of importance of extreme indicators is quite similar in North America, Europe and Asia. One notable distinction is the higher importance of Rx5day in Asian grid points compared to North America and Europe. The consistent selection of similar predictors across large spatial scales may suggest that the Lasso regression is fairly robust. However, this may also be related to the inevitable simplifications of crop growing processes in the

employed crop model. In particular, the same model is applied at all locations likely creating certain homogeneity by default. Kern et al. (2018) conducted a comparable analysis on observed winter wheat crop yield in Hungary. With a linear regression using a step-wise selection of monthly meteorological variables, they found that positive anomaly in VPD and $T_{min}$ during May decrease yield. Additionally, April, May and June appear to be the most relevant months in our global analysis, which is consistent with regional studies (Kern et al., 2018; Kogan et al., 2013; Ribeiro et al., 2020).

The occurrence of an extreme weather event in a given year may induce crop failure, underlining its importance as a predictor for crop failure. However, in years without such extreme events, crop yield are still governed by the weather during the growing season (Iizumi and Ramankutty, 2015). We found that both climate extreme indicators as well as monthly means of common climate variables have proven to be valuable predictors of years resulting in crop failure. Droughts and heat waves are well known to affect crop yield (Lesk et al., 2016; Jagadish et al., 2014), and temperature and precipitation explain a large fraction

of interannual crop yield variability (Lobell and Burke, 2008). In contrast, VPD is often overlooked statistical analyses of crop yield variability (Zhang et al., 2017). We show that VPD is a key predictor for crop failure. It is known to play a crucial role in plant functioning and is projected to increase as main limiting driver in the face of climate change (Novick et al., 2016; Grossiord et al., 2020). High VPD values can lead to plant mortality via carbon starvation and hydraulic failure (McDowell





et al., 2011; Cochard, 2019; Grossiord et al., 2020). However, its covariation with temperature and solar radiation makes it
difficult to disentangle their respective effects (Stocker et al., 2019; Grossiord et al., 2020). There are well-defined temperature
thresholds for wheat, e.g. a temperature of 31 °C before flowering is considered to evoke sterile grains and thus reducing
yield (Porter and Gawith, 1999; Daryanto et al., 2016). $T_{max}$ is a secondary predictor in our statistical model, which is in line
with results based on observed and simulated yields (Schauberger et al., 2017), and can be attributed to the rare exceedance
of critical temperature thresholds in the growing season. Crops are particularly vulnerable during key development stages, so
extreme events during that time span can lead to large yield reductions, even in case of otherwise favorable weather conditions
during the growing season (Porter and Gawith, 1999; Moriondo and Bindi, 2007). The vulnerability of wheat to climatic events
depends largely on phenophases, and generally wheat possesses a higher sensitivity to temperature and precipitation during
its reproductive phase than during its vegetative phase (Porter and Gawith, 1999; Luo, 2011; Daryanto et al., 2016). Future
research could investigate the importance of time of occurrence of extreme indicators (Vogel et al., 2019). For instance, due to
climate change false spring events may become more likely in some regions (Moriondo and Bindi, 2007; Allstadt et al., 2015)
and thus the timing of frost days could provide a valuable addition to the model.

The frequent inclusion of the extreme indicators dtr and frs in our regression model highlights that short-term extreme events
can potentially have larger impacts than gradual changes over time (Jentsch et al., 2007). The variable dtr was also identified
as an important predictor by Vogel et al. (2019), whereas frs was of minor importance for explaining variation in spring wheat
yield. By contrast, frs is one of the most predominant predictors in our study, which might be explained by the differing growing
season of winter wheat, which is encompassing primarily the cold seasons.

We explored the relevance of interactions between predictors, however this did not significantly improve model performance.
This might hint at the inability of the crop model APSIM to account adequately for such compound effects, which is consis-
tent with Ben-Ari et al. (2018), who linked the crop failure 2016 in France to an extraordinary combination of warm winter
temperatures followed by wet spring conditions. The commonly used crops models employed for crop yield forecasts were not
able to predict the 2016 yield failure in France (Ben-Ari et al., 2018).

Overall, our results illustrate the omnipresence of compounding meteorological events for crop failure. In nearly all grid
points, most seasons and meteorological variables were relevant to predict years with crop failure (Fig. 7). This suggests that
the co-occurrence of certain weather conditions as well as the combination of weather conditions in different seasons are
associated with crop failure. With our approach with have identified meteorological conditions that are statistically linked to
crop failure. To identify causal relationships, more advanced methods from the emerging field of causal inference could by
employed (Runge et al., 2019).

## 5 Conclusions

In this paper, we presented a robust statistical approach – namely Lasso logistic regression – for predicting crop failure and
automatically selecting relevant predictors among a large number of meteorological variables and climate extreme indicators.
We illustrated our approach on 1600 years of simulated winter wheat yield for the Northern Hemisphere under present-day



climate conditions. Lasso regression can serve as a tool for identifying important variables with automated variable selection, while accounting for collinearity and achieving overall good predictive power. Consistent with earlier knowledge, we find that predicting crop failure requires accounting for a number of different meteorological drivers at different times of the growing
season, which is illustrated by the large amount of variables at all seasons included in our statistical model (Fig. 7). This indicates that compounding effects are ubiquitous across time and meteorological drivers, and highlights the usefulness of approaches such as Lasso regression to reveal multiple meteorological drivers of crop failure. We identified vapour pressure deficit as one key variable to predict crop failure, which underlines the importance of its consideration in statistical crop yield models, in particular because it is often overlooked in statistical analyses of crop yield variability. Furthermore, climate extreme
indicators such as diurnal temperature range and the number of frost days have proven to be valuable additions to the predictive models, highlighting the necessity to address not only monthly mean conditions, but especially also climatic extremes in such models. Overall this study helps to enhance the knowledge required to improve seasonal forecasts and undertake adaptation measures against crop failure. The flexibility of our approach allows an application to other climate impacts that are influenced by a large range of variables varying with seasonality, for instance wildfires or flooding.

*Code and data availability.* The code to reproduce the figures is available from GitHub (https://github.com/jo-vogel/Identify_crop_yield_ drivers). The climate and crop simulations are available from Tianyi Zhang (zhangty@post.iap.ac.cn) and Karin van der Wiel (wiel@knmi.nl) on request, respectively.

*Video supplement.* Grid points with inclusion of monthly predictors for a) VPD, b) $T_{max}$ and c) Pr in the respective Lasso logistic regression model. The extension "Y1" means that the respective month belongs to the first calendar year of the growing season, while "Y2" means it
belongs to the second calendar year of the growing season.

**Appendix A: Additional figures**

*Author contributions.* J.Z. and K.v.d.W. conceived the project and supervised the work. J.V. and P.R. conducted most of the data analysis, including the Lasso logistic regression and creation of the key figures. K.v.d.W. performed the climate model simulations with EC-Earth. T.Z. performed the crop model simulations with APSIM. All authors contributed substantially to the data analysis, design of figures and writing
of the manuscript.

*Competing interests.* The authors declare that they have no competing interests.

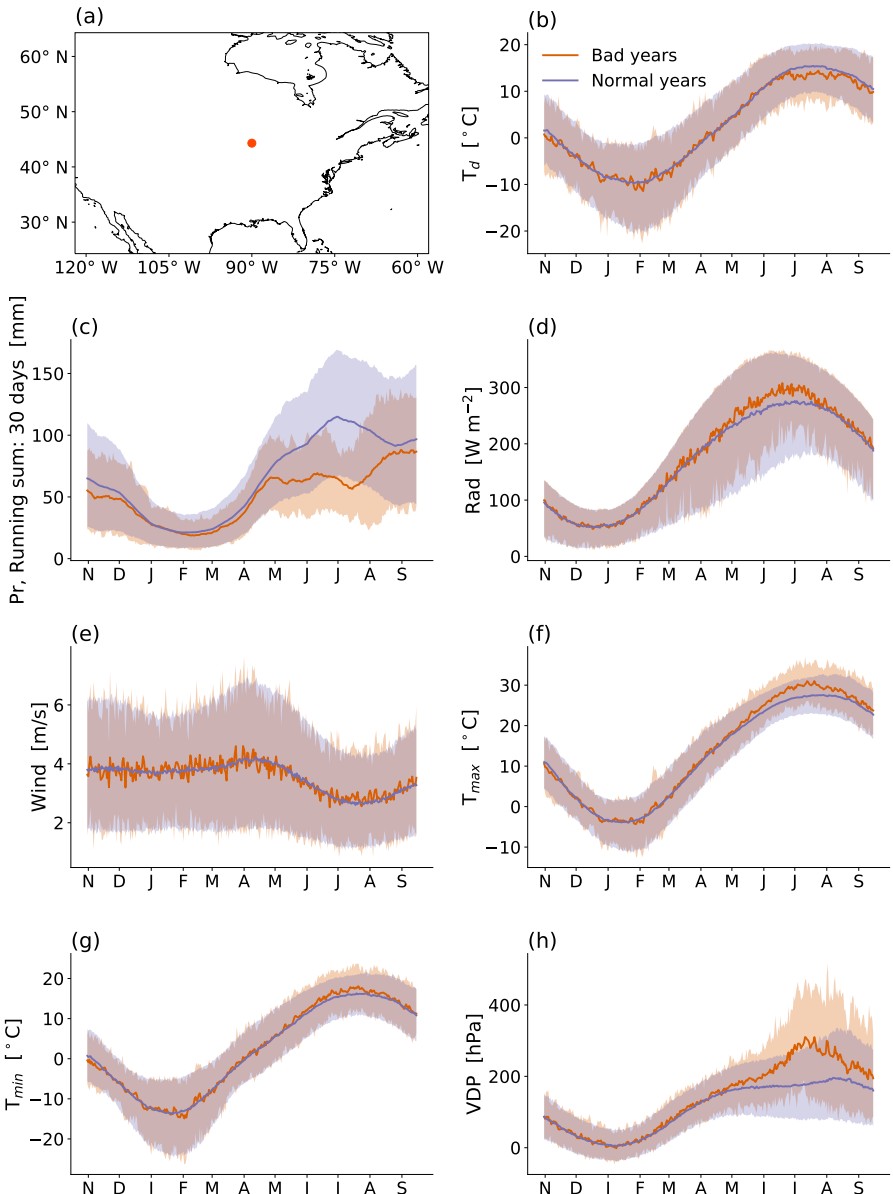

**Figure A1.** As Figure 2, but for a grid point in the United States (90.0° W, 44.3° N).

*Acknowledgements.*  This work emerged from the Training School on Statistical Modelling organized by the European COST Action DAMO-
CLES (CA17109). J.V. acknowledges funding by the DFG research training group "Natural Hazards and Risks in a Changing World" (Na-
tRiskChange GRK 2043). P.R. acknowledges funding from the the Swiss National Science Foundation (grant number 178751). C.A.S. is
funded by an EPSRC Doctoral Training Partnership (DTP) Grant (EP/R513349/1). E.T. and J.Z. acknowledge funding from the Swiss Na-
tional Science Foundation (Ambizione grant 179876). K.v.d.W. and T.Z. acknowledge funding for the HiWAVES3 project, funding was



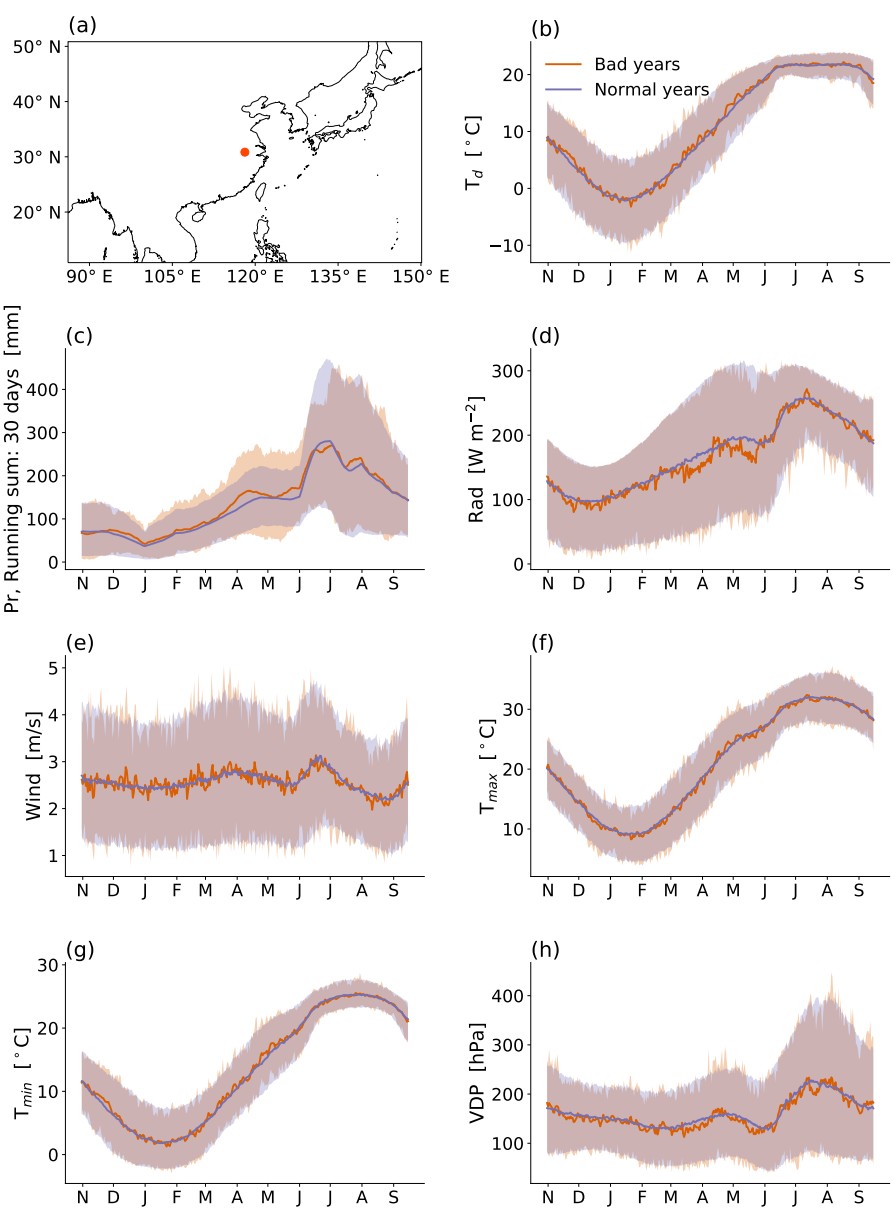

**Figure A2.** As Figure 2, but for a grid point in China (118.1° E, 30.8° N).

supplied through JPI Climate and the Belmont Forum (NWO ALWCL.2 016.2 and NSFC 41661144006), T.Z. further acknowledges the
National Key Research and Development Project of China (2019YFA0607402).



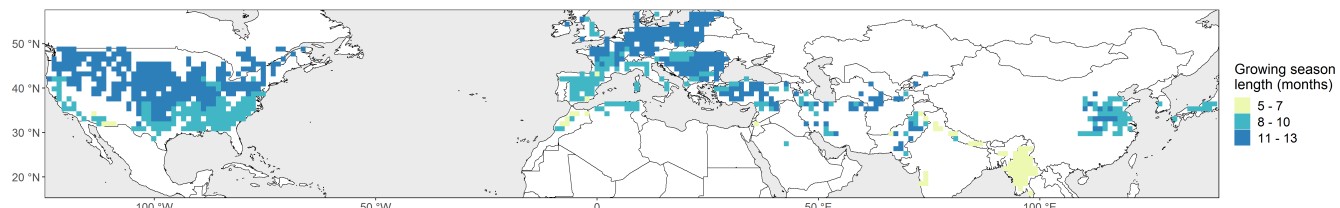

**Figure A3.** Number of months in the growing season (number of months between the earliest sowing date and the latest harvest date). The growing season starts at the month containing the sowing date and ends with the month containing the latest harvest date, among the 1600 model years. We discarded years with harvest date later than 365 days after the sowing date. Some growing seasons are 13 months long because we include both the entire first month and the entire last month.

**Figure A4.** Selected climate extreme indicators (Table 1) in the Lasso logistic regression model for each location. Diurnal temperature range (dtr, a), number of frost days (frs, b), TX90p (c) and Rx5day (d).



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
