# Peer review of "Identifying meteorological drivers of extreme impacts: an application to simulated crop yields"

_Earth System Dynamics, 2020_

## Referee Comment (RC1) · Anonymous Referee #1 · 21 Aug 2020

Vogel et al. showcase how a regression models for regional crop failures can be obtain from a multitude of potential climatological drivers while minimizing the number of relevant variables via LASSO, a statistical method. It's a nicely and thoroughly done analysis that deserves publication after some minor revisions.

**Minor Comments: 1.l. 3 understanding and forecasting of..?**

2.l. 8. 'predict' -> 'determine'?

3.l. 31 'depend'

4.l. 85 '..the climatology was defined to be the mean plus the first three annual har-

monics.' Can the authors further explain what is meant by that?

5.Figure 5 what is the growing season (GS) here could this possibly highlighted in the section to the left, which shows the months?

6.l. 167 and later on: Nice to see R-packages explicitly cited.

7.l. 190 is s segregation threshold and the local cut-off value? Maybe it would be better to use one term only?

8.Figure 7 sub-panels are not enumerated.

**Major Comments: 9.Could the authors provide an estimate on how many data points would be necessary at minimum to apply the LASSO method? Is there a relationship between total number of suggested variables and necessary datapoints?**

10.Do I understand correctly that LASSO, avoids autocorrelation by checking for the variability of variables only? Could the authors provide evidence for the reliability of such approach?

11.As prediction is mentioned as one application in the abstract, how would the CSI change if growing season variables were not included? In what region is it possible to build a useful prediction model using Months outside the growing season only, in what regions are yields not predictable without including conditions from within the Growing season itself

12.Overall but specifically in Fig.8 It might be more insightful to show results relative to the grid-point dependent growing season? Could this explain the differences in the shape of the histogram between the North America / Europe and Asia?

13.On data sampling: The 5th percentile is a rather low threshold, to increase the number of events (and come closer to a real-world applicability) is the method sensitive to the exact choice? What production loss does a 5th percentile correspond to regionally/on average?

---

## Referee Comment (RC2) · 13 Sep 2020

The paper is of a great interest and the authors applied an innovative approach to identify the drivers of extreme impact on crop yield. I found a few main issues that, on my opinion should be addressed before publication. The description of meteorological dataset is unclear and requires to be reframed under a more general scheme, e.g. which is the aim of developing such a dataset? Do you want to address the issue of uncertainties in climate simulations? Why using 1° as spatial resolution? May it be considered a good compromise between the scale you require for your assessment (global) and the scale required for crop growth simulations (local)? There is a reference

concerning the development of such dataset, but what is reported here is too much squeezed. The same applied for APSIM description. Given the topics of this journal, I strongly suspect that a general introduction about crop modelling is due. I would therefore expect a general overview on APSIM model with a particular reference to crop phenology, dry matter accumulation and limiting factors for growth. The effect of higher temperature on the length of growing season may therefore better understood as well as the effect of abiotic stresses on crop growth. As an example, this paper deals with the impact of extreme events on crop growth, but the reader actually does not understand which meteorological extremes can affect crop growth and if/how the model considers these impacts in relation the phenological stage. F Accordingly, I suggest firstly to outline the main features of APSIM. As a second step, I would state which are the main abiotic factors affecting crop yield with a special reference the phenological stage when they occur. E.g. a frost or heat events have a different impact if occurring during vegetative or reproductive stage. The authors report the issue on par 305, but the basis of this statement must be explained before by a description of impacts of extremes events in relation to crop growth Thirdly, I would explain how these effects are simulated in your model. E.g. how the impact of heat events at anthesis are simulated? Is there an additional effect during grain filling? This would help to better explain some trends observed in your results in relation the growing season period. This is of course a suggested scheme, but these issue should be in any case addressed Figure 1. Is there a general relationship between crop yield simulated and observed on the global scale? On par 320 there is a note on possible bias in crop growth calibration, but actually we do not know how crop model was calibrated The discussion tackles the effect of temporal resolution of the meteorological data and this an added value to the paper. In case, some discussion is due for some process not specifically considered in crop modelling approach

---

## Referee Comment (RC3) · Flavio Pons (Referee) · 4 Nov 2020

**1   Overview and major comment**

The paper is concerned with a relevant problem in a two-fold way: developing a methodology to detect which meteorological variables drive impacts on a certain socioeconomic variable, and giving an in-deep practical application to bad crop years.

I found the paper overall well written, and the problem is clearly stated and understandable even to a reader not familiar with crop modelling. The crop model remains quite a black box to the reader, but I feel like the main point of the article is to propose a

methodology and show its performance, rather than focusing on technical details of crop simulation.

The methodology itself is simply based on the application of a logistic lasso regression: the model is fed meteorological variable as covariates, to predict whether the yield at a certain grid point will result in a bad crop year (Y=0) or a good crop year (Y=1). The 'lasso' formulation allows to include a large number of predictors (in some cases even larger than the sample size) and only select - or, in some of its variants, group - a subset of predictors that reduces the problem dimensionality while maximizing forecasting performance. The model is tested against two competitors, a generalized linear model (I suppose binomial with logistic link, it would be nice to specify this detail in Section 2.5) and a random forest run in binary classification mode. The authors find comparable performances between lasso regression and random forest, however the latter is way less interpretable, making lasso a feasible yet effective way to model impact drivers.

As a major comment, which doesn't necessarily imply need for major revisions in the paper, I would like to stress that this greater interpretability is still quite limited by the nature of the lasso model. This is designed to select the variables that produce the best forecasting performance with minimal number of covariates in a linear model that may be a strong approximation of the real world phenomenon. This means that the selected variables are surely the ones that provide better explanation of crop failure *in the considered crop simulation model* and *in terms of prediction*. This does not necessarily imply selecting variables that directly physically drive the crop failure, just like the resulting regression coefficients are not estimates of a real linear law existing in nature, but of an approximation that optimizes forecasting.

In all fairness, results in the presented case study appear to be physically reasonable, and I found the discussion in Section 3.2 convincing in this sense. However, it is possible that in different problems, where processes are less understood, results can provide indications useful for forecasting but not really provide physical insights,

making the methodology not necessarily effective in all fields of application. I would explicitly stress this in the main body and in the conclusions, because a reader not familiar with the shortcomings of applied statistical modelling may over-generalise these findings to a problem where it is not possible to do so. Also, I would add a warning that critical interpretation of the results is always necessary, especially in cases with smaller or non gridded datasets, where the hints coming from spatial coherence (which in this paper play a role in making results more solid) may not be available.

**2 Minor/technical comments**

A general consideration: the notation calling "positive" years with a good crop may be a bit confusing when trying to interpret results. While a good yield is surely positive news, the model is designed to detect drivers of impacts leading to bad years: it would be more coherent with traditional terminology to address the non-baseline case under investigation with this term. I do not think that this is worth modifying the phrasing in the whole article, but maybe I would stress this, especially readers with a statistical rather/other than physical background may not pick up on this immediately (I didn't!).

1. (line 14) "both between" should read "of both"

2. (line 115) the authors state that they normalize all the variables to be in [-1,1]. I understand rescaling/normalizing variables when they take values that differ by several orders of magnitude, but I do not understand the choice of squeezing them into a close interval, as logistic regression handles continuous real valued covariates.

3. (line 150) the authors state that lasso is superior in handling correlations in the covariates better than standard GLMs. This is certainly true for correlation among
covariates, but I am not so sure about autocorrelation. In particular, meteorological data display a strong seasonality, which introduces long range autocorrelation in the data. Can the author provide some reference specific to this aspect?

4. (lines 168-175) I am not sure if I understand correctly the choice of $\lambda_{1se}$: is it because, using $\lambda_{min}+1se$ falls almost exactly in the middle of the 95% confidence interval that would require $2se$? If so, it makes sense but it should be explained more explicitly.

5. it seems that the authors choose a priori $s^* = 5\%$ and try also 2.5 and 10% to test the sensitivity as a threshold to define bad crop years. If so, does it make sense to define $s^*$ as the argmin of C(s) as in line 205?

6. (lines 219-222) not sure about these lines: it is a good idea to check for significant interactions and report it, but then I would explain in larger detail what interactions are in regression models, because the reader may not be familiar with the concept. Also, which one did they try, and did they have an a priori idea about possible meaningful interactions?

7. (line 231) "eastward" $\longrightarrow$ "westward"?

8. (line 327) the authors say that their analysis is based on a time series model, but maybe they mean that the dataset is constituted by gridded time series data.

9. (line 380) "With our approach with" should be "With our approach we"
* * *

---

## Author Comment (AC1) · 18 Nov 2020

We would like to thank the reviewers for their helpful and extensive comments, which further improved the article.

**Reviewer 1**

Vogel et al. showcase how a regression models for regional crop failures can be obtain from a multitude of potential climatological drivers while minimizing the number of relevant variables via LASSO, a statistical method. It's a nicely and thoroughly done analysis that deserves publication after some minor revisions.

**#### Minor Comments:**

1.l. 3 understanding and forecasting of..?

We will clarify the sentence:
*"Identifying the underlying mechanisms that cause extreme impacts, such as crop failure, is of crucial importance to improve their understanding and forecasting."*

2.l. 8. 'predict' -> 'determine'?

We will adjust the text accordingly.

3.l. 31 'depend'

We will adjust the text accordingly.

4.l. 85 '..the climatology was defined to be the mean plus the first three annual harmonics.' Can the authors further explain what is meant by that?

Harmonic analysis is a branch of mathematics which uses wave functions to describe data. What is meant by this statement is that we calculate the annual cycle by using harmonic analysis, and that we limit the mathematical description of this cycle to the first three wave functions.

In the figure you can see that this indeed captures the annual cycle and removes 'noise' due to weather.

[Figure]

5.Figure 5 what is the growing season (GS) here could this possibly highlighted in the section to the left, which shows the months?

Thank you for this helpful comment. We assume the reviewer meant figure 3, rather than figure 5. We will add the following sentences.

At the end of the caption of figure 3:
*"Note that a) shows the correlation for all months included in the growing season of the grid point in France and c) shows the average correlation for a given month computed over all grid points containing this month in their growing season"*

We will put a definition of growing season just before the introduction of Figure 2 in section 2.3:
*"For a given grid point, the sowing date is the same for the 1600 simulated years, but the harvest dates differ. We therefore define the growing season for a given grid point as starting on the month containing the sowing date and finishing with the month containing the latest harvest date."*

We will complete the last sentence of section 2.3:
*"We use monthly means of Tmax, Pr and VPD during the growing season, as well as the seven extreme indicators for further analysis."*

Figures 2, A2 and A3 will be adjusted, so that they now display the meteorological conditions starting from the sowing date of the corresponding location until the end of their longest growing season.

6. l. 167 and later on: Nice to see R-packages explicitly cited.

Thank you for the remark.

7. l. 190 is s segregation threshold and the local cut-off value? Maybe it would be better to use one term only?

Thank you for this remark. We chose to use exclusively the term segregation threshold in the revision.

8. Figure 7 sub-panels are not enumerated.

Thank you, we will correct this.

**Major Comments:**

9. Could the authors provide an estimate on how many data points would be necessary at minimum to apply the LASSO method? Is there a relationship between total number of suggested variables and necessary datapoints?

The user's guide of glmnet R-package recommends to apply the Lasso logistic regression only to a dataset containing more than 8 occurrences of each "1" and "0". As a consequence of the small number of bad years, defined here as years with yield below the 5$^{th}$ percentile, the dataset is more likely to have less than 8 bad years in the testing data with decreasing sample size. This results in a decreasing mean CSI with decreasing number of datapoints (see the figures below, presenting the CSI for Lasso regression applied with different configurations of the number of datapoints). We would therefore strongly recommend to have at least 8 occurrences of "1"s and 8 occurrences of "0"s in both the training and testing dataset, and a number of variables lower than the number of years available.

[Figure]

10. Do I understand correctly that LASSO, avoids autocorrelation by checking for the variability of variables only? Could the authors provide evidence for the reliability of such approach?

Thank you for your question. The advantage of Lasso we wanted to highlight is its ability to deal with potential correlation between explanatory variables. An example causing correlation is the autocorrelation within the annual time series of meteorological variables. As an example, the time series of temperature over the year presents autocorrelation due to seasonality, reflected in our case by correlation between e.g. the two explanatory variables "mean temperature in June" and "mean temperature in July". To avoid any confusion, we will remove the term "autocorrelation" in section 2.4.

Correlation between explanatory variables can be a problem in basic regression models. As explained in section 2.4, if two variables $X_1$ and $X_2$ are highly correlated, the information brought by a small regression coefficient ($beta_1$) for the variable $X_1$ and a large regression coefficient ($beta_2$) for the variable $X_2$ is the same as the information brought by a large $beta_1$ and a small $beta_2$. This implies large variability of the coefficients in the regression procedure. Lasso regression controls this variability of the regression coefficients with a penalty term on the norm of these coefficients.

The reader can find complete information about the reliability of the method regarding correlation between variables in Tibshirani (1996). We can provide here a short example with two correlated variables:

Tibshirani, R.: Regression Shrinkage and Selection via the Lasso, Journal of the Royal Statistical Society, 58, 267–288, 1996.

We can give a basic example of how the Lasso variable deals with correlation between two explanatory variables. Let $X_1$ and $X_2$ be two variables highly correlated, for example $X_2 = cX_1$ with $c > 1$. We would like to explain the yield variable $Y$ with only the two variables $X_1$ and $X_2$. Then the logistic Lasso regression consists in finding the minimum:

$$\min_{(\beta_0,\beta_1,\beta_2)\in\mathbb{R}^3} -\left[\frac{1}{n}\sum_{i=1}^{n} y_i \cdot (\beta_0 + \beta_1 X_1 + \beta_2 X_2) - \log(1 + e^{\beta_0+\beta_1 X_1+\beta_2 X_2})\right] + \lambda||\beta||_1$$

Let us rewrite $\mathcal{L}(Y, \beta_1 X_1 + \beta_2 X_2)$ the loss function in square brackets. The regression coefficients verify:

$$\hat{\beta} = \underset{(\beta_1,\beta_2)\in\mathbb{R}^2}{\operatorname{argmin}} \{-\mathcal{L}(Y, \beta_1 X_1 + \beta_2 X_2) + \lambda||\beta||_1\}$$

i.e.

$$\hat{\beta} = \underset{(\beta_1,\beta_2)\in\mathbb{R}^2}{\operatorname{argmin}} \{-\mathcal{L}(Y, (\beta_1 + c\beta_2)X_1) + \lambda(|\beta_1| + |\beta_2|)\}$$

i.e.

$$\hat{\beta} = \underset{k\in\mathbb{R}}{\operatorname{argmin}} \ \underset{(\beta_1,\beta_2)\in\mathbb{R}^2|\beta_1+c\beta_2=k}{\min} \{-\mathcal{L}(Y, kX_1) + \lambda(|\beta_1| + |\beta_2|)\}$$

i.e.

$$\hat{\beta} = \underset{k\in\mathbb{R}}{\operatorname{argmin}} \left\{-\mathcal{L}(Y, kX_1) + \underset{(\beta_1,\beta_2)\in\mathbb{R}^2,\beta_1+c\beta_2=k}{\min} \lambda(|\beta_1| + |\beta_2|)\right\}$$

It can be shown that the minimum on the $\beta$ coefficients is reached for $\beta_1 = 0$ (because $c > 1$), implying that the Lasso regression set to zero the contribution of $X_1$ to $Y$, as the information is already fully provided by $X_2$.

(example inspired by:
https://stats.stackexchange.com/questions/241471/if-multi-collinearity-is-high-would-lasso-coefficients-shrink-to-0 )

11. As prediction is mentioned as one application in the abstract, how would the CSI change if growing season variables were not included? In what region is it possible to build a useful prediction model using Months outside the growing season only, in what regions are yields not predictable without including conditions from within the Growing season itself

Thank you for pointing out this potential application of our work.
We do not apply Lasso regression for forecasting, as this is beyond the scope of our study.
In principle, the set of selected variables by the Lasso regression at each grid point could be used as a starting point to identify regions, where parts of the growing season could potentially be excluded and to assess predictive power for forecasts in future work.
We also would like to clarify that our study is designed to include only months belonging to the maximum growing season of each pixel, whereas prior months before the growing season are not taken into account here.

12. Overall but specifically in Fig.8 It might be more insightful to show results relative to the grid-point dependent growing season? Could this explain the differences in the shape of the histogram between the North America / Europe and Asia?

Thank you for this comment. We were also considering this option (see the corresponding figure attached below). However, it does not add substantial additional value in our opinion and either way it is a trade-off.
If we show which month of the growing season relative to the grid-point (Month 1, 2, 3,...) is selected, we lose the information which month of year (January - December) is selected and vice versa. Furthermore, if we use the sowing date of each grid point as a base line, we still cannot infer the precise stage of the growing season because the harvest dates also differ (and therefore the length of the growing season).
Yes, the differences in the shapes of the histograms can likely be explained by differences in the start and length of the growing season. This is addressed at the end of the second paragraph in section 3.2.

[Figure]

Figure 8. For each possible predictor we show the percentage of grid points for which this predictors has a non-zero coefficient in the Lasso logistic regression. (a) all continents (889 grid points in total), (b) North America (419 grid points), (c) Europe (233 grid points) and (d) Asia (210 grid points). The numbers at the end indicate the month of the growing season according to the respective sowing date of each grid point.

13.On data sampling: The 5th percentile is a rather low threshold, to increase the number of events (and come closer to a real-world applicability) is the method sensitive to the exact choice? What production loss does a 5th percentile correspond to regionally/on average?

*What production loss does a 5th percentile correspond to regionally/on average?*
Thank you for your question. We will complete figure 1 with a map of the relative difference of the 5th percentile threshold and the mean yield. For a majority of grid points, the 5th percentile value is between 30 and 60% of the mean yield. One can note that some regions with low CSI are also regions with a small relative difference between the mean yield and the 5th percentile threshold, e.g. in southern China and Japan. Low yield variability can therefore lead to low model performance, suggesting a challenging distinction between normal and bad years.

[Figure]

*Is the method sensitive to the exact choice?*
To address the sensitivity towards the chosen threshold for crop failure, we performed the analysis additionally for the 10th percentile. We attached reproduction of figures 5, 6, 7, 8 and A4 in the main manuscript for the 10th percentile below. In general, these results are very similar to the 5th percentile, thus our approach seems to be not very sensitive to the choice of the percentile. The CSI generally increases a bit (the mean CSI is 0.52), which indicates the distinction between crop failure and normal years becomes better with an increasing threshold. This improvement is likely due to the higher amount of data assigned to extreme crop yield loss. This shows that data availability is crucial for good model performance. In our study, we decided for the 5th percentile as a trade-off between data availability and the magnitude of the extreme (which is the focus of our research).

[Figure]

Figure 5

[Figure]

Figure 6

[Figure]

Figure 7

[Figure]

Figure 8

[Figure]

Figure A4

**Reviewer 2**

The paper is of a great interest and the authors applied an innovative approach to identify the drivers of extreme impact on crop yield. I found a few main issues that, on my opinion should be addressed before publication.

The description of the meteorological dataset is unclear and requires to be reframed under a more general scheme, e.g. which is the aim of developing such a dataset? Do you want to address the issue of uncertainties in climate simulations? Why using 1 degree as spatial resolution? May it be considered a good compromise between the scale you require for your assessment (global) and the scale required for crop growth simulations (local)? There is a reference concerning the development of such a dataset, but what is reported here is too much squeezed.

The large ensemble climate model experiment was developed to investigate natural variability and extreme events in the climate system, and their influence on societal/natural impacts. Creating large ensemble simulations is computationally very expensive, hence the horizontal resolution generally remains relatively low, this is indeed a compromise as the reviewer correctly notes. We note however, that these simulations are state-of-the-art in the climate modelling community, comparable in ensemble size (here 2000 years) and horizontal resolution (here ~1 degree) to other efforts (e.g. MMLEA at http://www.cesm.ucar.edu/projects/community-projects/MMLEA/).

All climate models are subject to possible model biases. However, the climate model simulations were designed to match observed global mean temperatures between 2011-2015. Since the aim of the present paper is mostly methodological, i.e. can we identify drivers of extreme impacts using Lasso regression, we do not think this is an issue here.
We will rewrite the paragraph to explain the purpose of the climate model simulations, and note the compromise between spatial resolution and ensemble size.
*"To investigate the influence of natural variability and climatic extreme events, a large ensemble simulation experiment was set up with the EC-Earth global climate model (v2.3, Hazeleger et al., 2012). We use this climate model data set, consisting of 2000 years of present-day simulated weather, to investigate if we can identify the drivers of extreme low crop yield seasons. Large ensemble modelling is at the forefront of climate science (Deser et al., 2020), due to the computational expenses involved a balance between ensemble size, horizontal resolution and number of climate models has to be found. We have found the climate data used here to be suitable for the presented study. A detailed description of these climate simulations is provided in Van der Wiel et al. (2019b), here we provide a short overview of the experimental setup. Present-day was defined as the five year model period in which the global mean surface temperature matched that observed in 2011-2015 (HadCRUT4 data, Morice et al., 2012). Because of a cold bias in EC-Earth, in the model this period is 2035-2039. To create the large ensemble, twenty five ensemble members were branched off from sixteen long transient climate runs (forced by Representative Concentration Pathway (RCP) 8.5). Each ensemble member was integrated for five years.*

*Differences between ensemble members were forced by choosing different seeds in the atmospheric stochastics perturbations (Buizza et al., 1999). This resulted in a total of 16 x 25 x 5 = 2000 years of meteorological data, at T159 horizontal resolution (approximately 1°)."*

Buizza, R., Milleer, M., and Palmer, T. N.: Stochastic representation of model uncertainties in the ECMWF ensemble prediction system, Quarterly Journal of the Royal Meteorological Society, 125, 2887–2908, 1999.

Deser, C., Lehner, F., Rodgers, K., Ault, T., Delworth, T., DiNezio, P., Fiore, A., Frankignoul, C., Fyfe, J., Horton, D., et al.: Insights from Earth system model initial-condition large ensembles and future prospects, Nature Climate Change, pp. 1–10, https://doi.org/10.1038/s41558-020-0731-2, 2020.

Hazeleger, W., Wang, X., Severijns, C., ̧ Stefanescu, S., Bintanja, R., Sterl, A., Wyser, K., Semmler, T., Yang, S., Van den Hurk, B., et al.: EC-Earth V2.2: description and validation of a new seamless earth system prediction model, Climate dynamics, 39, 2611–2629, 2012.

Morice, C. P., Kennedy, J. J., Rayner, N. A., and Jones, P. D.: Quantifying uncertainties in global and regional temperature change using an ensemble of observational estimates: The HadCRUT4 data set, Journal of Geophysical Research: Atmospheres, 117, 2012.

Van der Wiel, K.,Wanders, N., Selten, F., and Bierkens, M.: Added value of large ensemble simulations for assessing extreme river discharge in a 2°C warmer world, Geophysical Research Letters, 46, 2093–2102, 2019b.

The same applied for APSIM description. Given the topics of this journal, I strongly suspect that a general introduction about crop modelling is due. I would therefore expect a general overview on the APSIM model with a particular reference to crop phenology, dry matter accumulation and limiting factors for growth. The effect of higher temperature on the length of growing season may therefore be better understood as well as the effect of abiotic stresses on crop growth. As an example, this paper deals with the impact of extreme events on crop growth, but the reader actually does not understand which meteorological extremes can affect crop growth and if/how the model considers these impacts in relation to the phenological stage. F Accordingly, I suggest firstly to outline the main features of APSIM.

As a second step, I would state which are the main abiotic factors affecting crop yield with a special reference to the phenological stage when they occur. E.g. a frost or heat events have a different impact if occurring during vegetative or reproductive stage. The authors report the issue on par 305, but the basis of this statement must be explained before by a description of impacts of extremes events in relation to crop growth

Thirdly, I would explain how these effects are simulated in your model. E.g. how the impact of heat events at anthesis are simulated? Is there an additional effect during grain filling? This would help to better explain some trends observed in your results in relation to the growing season period. This is of course a suggested scheme, but these issue should be in any case addressed Figure 1. Is there a general relationship between crop yield simulated and observed on the global scale? On page 320 there is a note on possible bias in crop growth calibration, but actually we do not know how the crop model was calibrated. The discussion tackles the effect of temporal resolution of the meteorological data and this an added value to the paper. In case, some discussion is due for some process not specifically considered in crop modelling approach

As suggested, we will add an appendix to introduce the APSIM model. We generally follow the outlines that were given and introduce the phenology and biomass algorithm of the APSIM-Wheat model. The APSIM-Wheat model does not currently simulate the specific effects of heat or frost stress events on grain or floret sterility. Previous studies have

suggested that low yields are not necessarily provoked by weather extremes, but are sometimes rather caused by moderate climate conditions (van der Wiel et al., 2020). Therefore, this study is focusing on yield extremes resulting from weather conditions during the growing season, rather than climatic extreme events during specific phenological phases.

Besides model description, we will also add a figure validating the APSIM output against country yield statistics in the appendix (see figure below).

K van der Wiel, FM Selten, R Bintanja, R Blackport, JA Screen (2020): Ensemble climate-impact modelling: extreme impacts from moderate meteorological conditions. Environmental Research Letters, 15, pp. 034050.

[Figure]

**Reviewer 3**

**#### Major Comments:**

As a major comment, which doesn't necessarily imply need for major revisions in the paper, I would like to stress that this greater interpretability is still quite limited by the nature of the Lasso model. This is designed to select the variables that produce the best forecasting performance with minimal number of covariates in a linear model that may be a strong approximation of the real world phenomenon. This means that the selected variables are surely the ones that provide better explanation of crop failure in the considered crop simulation model and in terms of prediction. This does not necessarily imply selecting variables that directly physically drive the crop failure, just like the resulting regression coefficients are not estimates of a real linear law existing in nature, but of an approximation that optimizes forecasting.

In all fairness, results in the presented case study appear to be physically reasonable, and I found the discussion in Section 3.2 convincing in this sense. However, it is possible that in different problems, where processes are less understood, results can provide indications useful for forecasting but not really provide physical insights, making the methodology not necessarily effective in all fields of application. I would explicitly stress this in the main body and in the conclusions, because a reader not familiar with the shortcomings of applied statistical modelling may over-generalise these findings to a problem where it is not possible to do so. Also, I would add a warning that critical interpretation of the results is always necessary, especially in cases with smaller or non gridded datasets, where the hints coming from spatial coherence (which in this paper play a role in making results more solid) may not be available.

*Point 1: Interpretability of results*

The reviewer raises an important point, which is partially already addressed in the article. To avoid confusion and improve clarity, we will add additional text.

We highlight this point at the end of the abstract and in section 4.2 that the detected relationships are of purely correlative nature and thus do not necessarily imply a causal structure between drivers and impacts. In that sense, our article presents a method to identify potential relevant drivers, whose physical meaning could be investigated in a next step, e.g. by applying causal inference frameworks (Runge et al. 2019).

We will add a sentence in section 4.2 in line 381 (as penultimate of the paragraph) to make this point clearer to the reader:

*"However, for interpretation of the selected variable set one should be aware that the variables in our model are selected based on correlation, and thus attributing them as potential physical drivers needs further careful investigation."*

Runge, Jakob; Bathiany, Sebastian; Bollt, Erik; Camps-Valls, Gustau; Coumou, Dim; Deyle, Ethan et al. (2019): Inferring causation from time series in Earth system sciences. In *Nature communications* 10 (1), p. 2553. DOI: 10.1038/s41467-019-10105-3.

*Point 2: Importance of data size and spatial extent*

Using a sample size of 400 (a quarter of the available data) we still obtain an average CSI of 0.33, indicating that performance decreases only slightly with decreasing data size. For further details, please see the answer to question 9 in the first review.

Thank you for your remark on spatial coherence. We agree that observational data is often not available at such spatial extent as the crop data used here and therefore spatial coherence cannot be used as an indicator of robustness for observational data. We will add a few sentences on this in section 4.1:

*"We analysed the robustness of our results using a) the $10^{th}$ percentile as a threshold to discriminate between bad and normal years and b) a smaller data subset with only 400 entries per grid point (i.e. a quarter of the available data). The spatial patterns of the selected predictors and the CSI (results not shown) using a $10^{th}$ percentile threshold are very similar compared to those of the $5^{th}$ percentile and the average CSI increases slightly from 0.43 to 0.52. Using a sample size of 400 we still obtain an average CSI of 0.33 - indicating that performance decreases only slightly with decreasing data size -, while the spatial patterns remain consistent (results not shown). Furthermore, the spatial coherence of our results additionally points out the robustness of our analysis. An application of the approach on real data might still be challenging, as observational sample sizes generally are much smaller than even 400 years. In addition, observational data is often not available at such spatial extent as it is the case for the crop model data used in this study. This makes the analysis of spatial coherence - which can serve as an indicator of model robustness - unfeasible."*

**#### Minor Comments:**

The model is tested against two competitors, a generalized linear model (I suppose binomial with logistic link, it would be nice to specify this detail in Section 2.5) and a random forest run in binary classification mode.

Yes, it is binomial with a logit link function. We will add a remark on this in section 2.5:
*", using a binomial family with a logit link function."*

A general consideration: the notation calling "positive" years with a good crop may be a bit confusing when trying to interpret results. While a good yield is surely positive news, the model is designed to detect drivers of impacts leading to bad years: it would be more coherent with traditional terminology to address the non-baseline case under investigation with this term. I do not think that this is worth modifying the phrasing in the whole article, but maybe I would stress this, especially readers with a statistical rather/other than physical background may not pick up on this immediately (I didn't!).

*We agree that this might be a bit confusing in the current version and will therefore adjust the text accordingly and will refer to the "bad years" as "positives" in the classification.*

1. (line 14) "both between" should read "of both"

*We will adjust the text accordingly.*

2. (line 115) the authors state that they normalize all the variables to be in [-1,1]. I understand rescaling/normalizing variables when they take values that differ by several orders of magnitude, but I do not understand the choice of squeezing them into a close interval, as logistic regression handles continuous real valued covariates.

*We also used z-score standardization, which yielded the same results. Some of the variables have skewed distributions (e.g. some of the extreme indicators) so we decided that a normalization to the interval [-1,1] would be more appropriate.*

3. (line 150) the authors state that Lasso is superior in handling correlations in the covariates better than standard GLMs. This is certainly true for correlation among covariates, but I am not so sure about autocorrelation. In particular, meteorological data display a strong seasonality, which introduces long range autocorrelation in the data. Can the author provide some reference specific to this aspect?

*We refer to our answer to question 10 in the first review. Time series of meteorological variables are surely autocorrelated over the course of the year. However each month of the growing season is considered as a separate climate variable in our model: there is correlation among variables, but no autocorrelation within a variable, as years can be considered as independent. We will remove the word "autocorrelation" from the manuscript to avoid confusion.*

4. (lines 168-175) I am not sure if I understand correctly the choice of $\_{1se}$: is it because, using $\_{min}$+1se falls almost exactly in the middle of the 95% confidence interval that would require 2se? If so, it makes sense but it should be explained more explicitly.

*The choice of $\lambda_{1se}$ was motivated by a trade-off when minimizing both the errors produced by the model and the number of variables. We followed guidelines by the authors of the glmnet package (Friedman et al. 2010) and by Krstajic et al. 2014 ("The main point of the 1 SE rule, with which we agree, is to choose the simplest model whose accuracy is comparable with the best model"). We will add these references in the manuscript.*
Friedman, J., Hastie, T., and Tibshirani, R.: Regularization Paths for Generalized Linear Models via Coordinate Descent, Journal of Statistical Software, Articles, 33, 1–22, https://doi.org/10.18637/jss.v033.i01, 2010.
Krstajic, D., Buturovic, L. J., Leahy, D. E., and Thomas, S.: : Cross-validation pitfalls when selecting and assessing regression and classification models, Journal of Cheminformatics, 6, 1–15, https://doi.org/10.1186/1758-2946-6-10, 2014.

5. it seems that the authors choose a priori $s_* = 5\%$ and try also 2.5 and 10% to test the sensitivity as a threshold to define bad crop years. If so, does it make sense to define $s_*$ as the argmin of C(s) as in line 205?

This question concerns two different thresholds. The first one is the "percentile threshold" (in our case 5%), which is applied to discriminate between bad and normal yields. The second one is the "segregation threshold" s*, which is used to transform the continuous predictions of the Lasso regression to binary classes. We are aware that these two thresholds can potentially be confounded. To avoid ambiguity, we will strictly refer to them as "percentile threshold" and "segregation threshold" in the revised version, respectively.

6. (lines 219-222) not sure about these lines: it is a good idea to check for significant interactions and report it, but then I would explain in larger detail what interactions are in regression models, because the reader may not be familiar with the concept. Also, which one did they try, and did they have an a priori idea about possible meaningful interactions?

We investigated first order interactions between potential drivers. To avoid confusion, we will remove this paragraph in the revision.

7. (line 231) "eastward" → "westward"?

We will adjust the text accordingly.

8. (line 327) the authors say that their analysis is based on a time series model, but maybe they mean that the dataset is constituted by gridded time series data.

We agree that the term "time series model" might be a bit misleading in this context. We will rephrase this sentence as follows:

*"Our analysis was based on fitting a local model at each location, which is one of the three principal statistical methods used to link crop yield with weather conditions, along with cross section models and panel models, which are global models that adjust for spatial variability using fixed or random effects (Lobell & Burke, 2010; Shi et al., 2013)."*

Lobell, D. B. and Burke, M. B.: On the use of statistical models to predict crop yield responses to climate change, Agricultural and Forest Meteorology, 150, 1443–1452, https://doi.org/10.1016/j.agrformet.2010.07.008, 2010.
Shi, W., Tao, F., and Zhang, Z.: A review on statistical models for identifying climate contributions to crop yields, Journal of Geographical Sciences, 23, 567–576, https://doi.org/10.1007/s11442-013-1029-3, 2013.

9. (line 380) "With our approach with" should be "With our approach we"

We will adjust the text accordingly.

---

## Author Response (AR1)

We would like to thank the reviewers for their helpful and constructive comments, which, we believe, further improved the article. The major changes implemented in the manuscript include a more detailed description of the climate and the crop model simulations in section 2.1 and Appendix A and a paragraph on the robustness of the results of the Lasso regression in section 4.1

**Reviewer 1**

Vogel et al. showcase how a regression models for regional crop failures can be obtain from a multitude of potential climatological drivers while minimizing the number of relevant variables via LASSO, a statistical method. It's a nicely and thoroughly done analysis that deserves publication after some minor revisions.

**#### Minor Comments:**

1.I. 3 understanding and forecasting of ..?

**We clarified the sentence:**

*"Identifying the underlying mechanisms that cause extreme impacts, such as crop failure, is of crucial importance to improve their understanding and forecasting."*

2.I. 8. 'predict' -> 'determine'?

We adjusted the text accordingly.

**3.I. 31 'depend'**

We adjusted the text accordingly.

4.I. 85 '..the climatology was defined to be the mean plus the first three annual harmonics.' Can the authors further explain what is meant by that?

Harmonic analysis is a branch of mathematics which uses wave functions to describe data. What is meant by this statement is that we calculate the annual cycle by using harmonic analysis, and that we limit the mathematical description of this cycle to the first three wave functions.

In the figure you can see that this indeed captures the annual cycle and removes 'noise' due to weather.

5.Figure 5 what is the growing season (GS) here could this possibly highlighted in the section to the left, which shows the months?

Thank you for this helpful comment. We assume the reviewer meant figure 3, rather than figure 5. We added the following sentences.

**At the end of the caption of figure 3:**

"Note that (a) shows the correlation for all months included in the growing season of the grid point in France and (c) shows the average correlation for a given month computed over all grid points containing this month in their growing season."

We put a definition of growing season just before the introduction of Figure 2 in section 2.3: "For a given grid point, the sowing date is the same for the 1600 simulated years, but the harvest dates differ. We therefore define the growing season for a given grid point as starting on the month containing the sowing date and finishing with the month containing the latest harvest date."

**We completed the last sentence of section 2.3:**

"We use monthly means of Tmax, Pr and VPD during the growing season, as well as the seven extreme indicators for further analysis."

Figures 2, A2 and A3 were adjusted, so that they now display the meteorological conditions starting from the sowing date of the corresponding location until the end of their longest growing season.

6. I. 167 and later on: Nice to see R-packages explicitly cited.

Thank you for the remark.

**7. I. 190 is s segregation threshold and the local cut-off value? Maybe it would be better to use one term only?**

Thank you for this remark. We chose to use exclusively the term segregation threshold in the revision.

8. Figure 7 sub-panels are not enumerated.

Thank you, we corrected this.

**#### Major Comments:**

9. Could the authors provide an estimate on how many data points would be necessary at minimum to apply the LASSO method? Is there a relationship between total number of suggested variables and necessary datapoints?

The user's guide of glmnet R-package recommends to apply the Lasso logistic regression only to a dataset containing more than 8 occurrences of each "1" and "0". As a consequence of the small number of bad years, defined here as years with yield below the 5th percentile, the dataset is more likely to have less than 8 bad years in the testing data with decreasing sample size. This results in a decreasing mean CSI with decreasing number of datapoints (see the figures below, presenting the CSI for Lasso regression applied with different configurations of the number of datapoints). We would therefore strongly recommend to have at least 8 occurrences of "1"s and 8 occurrences of "0"s in both the training and testing dataset, and a number of variables lower than the number of years available. We added a remark on the robustness of our results with decreasing sample size (see major comment in the third review).

---

## Author Response (AR2)

**Second revision**

We addressed the minor technical edit suggested by Reviewer #1 in line 225 in our resubmitted manuscript main_second_revision.pdf:

"In addition to the normalization by rescaling the data to the interval [-1, 1], we also performed a z-score transformation, which yielded comparable results. Therefore our choice of normalization is arbitrary to a degree and a z-score transformation can potentially also be applied in the Lasso logistic regression model."